# What Difference Gender Makes: Obstetricians in Nineteenth-Century Brazil

Marcia Esteves Agostinho 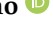

Department of History, University of Rochester, Rochester, NY 14627, USA; mesteves@ur.rochester.edu

**Abstract:** Like in most places around the world, childbirth assistance in Brazil was traditionally performed by women. In 1832, however, a law was passed requiring a license for the exercise of medicine, pharmacy, and midwifery. That event marked the differentiation between the traditional and the modern kind of childbirth assistants, leading to an increasing process of medicalization of birth. Hence, the historiography on the subject has pointed out the appropriation by men of a traditional women's world. This article seeks to understand the gender dynamics in the birthing room by focusing on the new kind of professional that emerged in Brazil in the early nineteenth century: the "graduated midwife." To what extent was there cooperation or competition between physicians and graduated midwives? How different were their obstetrical practices? After examining the *Annaes Brasiliensis de Medicina*—the official publication of the Imperial Academy of Medicine—I argue that the graduated midwife was the historical intermediate in transitioning from traditional midwifery to scientific obstetrics. Finally, I conclude that, as a woman of science, the graduated midwife filled the gap that isolated the female sphere of care from the male sphere of science, paving the road for the entrance of women in medicine in 1879.

**Keywords:** medicalization of birth; midwifery; women in medicine





## 1. Introduction

Obstetrics in Brazil has a date of birth: 18 February 1808. Threatened by the Napoleonic forces, the Portuguese Crown transferred to Brazil, bringing an extensive court along. It did not take long for Prince Regent D. João to realize the necessity of providing official medical education in the country that was to become his home for the next twelve years. Thus, in less than one month after their arrival, D. João signed a legal act founding the first medical schools in Brazilian territory. In addition to Surgery and Anatomy, the initial curriculum included Obstetrical Arts, showing that there was the intention to provide medical services to groups other than the armed forces, as some historians have assumed [1] (p. 47). That was the first step in the historical process of medicalization of childbirth in Brazil—a topic of continuing debate until today [2,3].

Like in most places around the world, childbirth assistance in Brazil was traditionally performed by women. Nineteenth-century traditional midwives played a "noninterventionist, supportive role" very similar to that described by Judith Walzer Leavitt regarding the United States. Her assertion that midwives traditionally "let nature take its course" is also valid for their Brazilian counterparts [4] (p. 38). This is evident in the language used in Brazil to refer to the women that assisted childbirth. They were known as "aparadeiras" [catchers]—implying they just waited to catch the child that was supposed to be naturally expelled by the woman's body—or "comadres" [godmother of one's child]—a term that suggests the support family members give. In both countries, when labor did not progress so naturally as expected, "midwives might have turned the fetus—"version"—or fortified women with hard liquor or mulled wine. They may have manually stretched the cervix or, rarely, administered ergot, a drug that stimulated contractions" [4] (p. 38).

However, in nineteenth-century Brazil, complicated cases were increasingly a prerogative of a new kind of professional: the "graduated midwife." In 1832, a law was passed

requiring a license for the exercise of Medicine, pharmacy, and midwifery [5]. Therefore, as the nineteenth century advanced, practices such as "version" and drugs administration were ever more expected from graduated midwives, stressing the increasing differentiation between the traditional and the modern kind of childbirth assistants. In this context, complicated births would make traditional midwives call a graduated midwife or a physician for help—at least in urban centers where trained professionals were available.

Leavitt argues that "the entrance of physician-accoucheurs into the practice of obstetrics in America during the second half of the eighteenth century marked the first significant break with tradition" [4] (p. 38). In Brazil, the entrance of male midwives would not happen until the early nineteenth century. Before that, the total number of physicians was too small to afford their time to be expended in activities that experienced women in the communities could perform. However, with the foundation of the Medical Schools of Rio de Janeiro and Bahia, both women and men began to receive systematic scientific training and certification on obstetrics. These institutions offered a two-year specialized course for women who wanted to become "graduated midwives." The same course was offered—in the same classroom—to senior male students pursuing a medical degree. So, for the male students, the midwifery course was the subject of the last two years of a five-year program that graduated Doctors in Medicine. Yet, some historians suggest that only a few physicians had some interest in practicing obstetrics [6].

Some themes stand out in the historiography on the medicalization of childbirth in Brazil. They include the development of scientific knowledge and medical specializations; the education, licensing, and performance of midwives; and the competition between graduated midwives and physicians. No matter what the theme is, historiography on this topic tends to echo the idea that "the inclusion of physicians-obstetricians in this practice led not only to a scrutinizing of the female body but also to the production of an anatomical and physiological knowledge of the female organism from a male standpoint" [7]. The arguments are based on the omnipresence of men in scientific and regulatory institutions and, consequently, in positions of power over women. Although many historians may not consider themselves feminist scholars, there is a gendered stance in most scholarly studies. Yet, few of them approach this matter using gender as an analytical tool. Instead of seeking to understand the gender dynamics of physicians' entry into the birthing room, authors that write about the medicalization of childbirth in Brazil seem to be satisfied with just using the activist rhetoric.

Decades ago, Joan Scott pointed out the capacity for historically constructed sexual roles to explain social dynamics [8]. Thus, the realization that "gender" carries knowledge about social interactions in historical contexts makes it a powerful analytical tool for those who want to understand the past. Yet, to identify gendered relations is not the same as understanding their meanings and functions. In each of the essays in her book, sometimes indirectly, Scott stresses the need to pose a fundamental historical question: "Since when?" When did certain differences start, or cease, to matter? In which circumstances has equality become a value in the modern world, and to what extent? The present research aims to contribute to this ongoing conversation by placing gender in the center of the methodology, which starts by reframing the research questions accordingly. Hence, I pose the following research questions. Considering that, in nineteenth-century Brazil, future male physicians and female graduated midwives were educated in the same obstetrics courses at the imperial medical schools, how different were their obstetrical practices? To what extent was there cooperation between physicians and graduated midwives instead of competition? What difference did their gender make in the delivery room, be it inside family homes or hospitals?

This research seeks to answer these questions by examining the official publication of the Imperial Academy of Medicine, the *Annaes Brasiliensis de Medicina*. The issues from 1851 to 1885 are available online in the Biblioteca Nacional digital archives [National Library of Brazil]. Despite the intrinsic importance of this periodical, a historical fact makes it particularly relevant as a primary source for this research. The first midwife to graduate

in Brazil (in 1834) let her words for posterity on the pages of the *Annaes.* Her name was Marie Josefina Mathilde Durocher (1809–1893), the midwife of the Imperial House who was also the only female member of the Academy of Medicine and a prolific contributor to its periodical [9] (p. 241). During her almost sixty years of practice, Mme. Durocher assisted more than 5000 deliveries. Although her biographer claims that Durocher's life "is similar to that of other women in urban Brazil in the nineteenth century who practiced the same profession of obstetrics" [10], she is the only one to publish detailed cases reports. To better make sense of the transformations in obstetrical practices in nineteenth-century Brazil, I analyze Durocher's texts in parallel with the writings of other obstetricians of the time. They were another graduated midwife named Ms. Tygna, and two male physicians, Dr. Gama Lobo and Dr. Vicente Saboia. The first was editor of the *Annaes*, and the latter would become the Director of the Medical School of Rio de Janeiro when women were accepted as full-students in the medicine course in 1879.

Decades of medical reports reveal how the process of medicalization of childbirth weaved the trajectories of these three kinds of assistants: the traditional midwife, the graduated midwife, and the physician. While the traditional midwife represented an ancient craft, the physician, on the opposite side, embodied the nineteenth-century quest for its scientific legitimation. The graduated midwife emerged from the tension between tradition and modernity in healthcare and other spheres of life. Thus, I argue that the graduated midwife in Brazil was the historical 'intermediate' in the transition from traditional midwifery to scientific obstetrics during the modernizing nineteenth century. In the natural sciences, "an intermediate or reaction intermediate is a substance formed during a middle step of a chemical reaction between reactants and the desired product. Intermediates tend to be extremely reactive and short-lived, so they represent a low concentration in a chemical reaction compared with the amount of reactants or products. Many intermediates are unstable ions or free radicals" [11]. Like an intermediate substance that disappears once the chemical reaction is complete, this historical intermediate has its ephemeral existence justified by serving to transform midwives into female physicians. A reform established by law in 1925 eliminated the midwifery course and created a course for nurses in maternity wards attached to the medical schools [12]. Despite having existed for less than one hundred years in Brazil, I show how graduated midwives built a bridge for women to enter the medical profession. Through their history, it is possible to infer the difference that gender may—or may not—have made in Brazilian obstetrics.

Before proceeding, it is worth noting that, unlike other linguistic contexts, the Portuguese word analogous to "obstetrics" [*obstetrícia*] was already commonly used in the early 1800s [13]. Note that the *Jornal de Coimbra* (1812) advertised "*the Quincy's Lexicom Medicon*" translating the English word "midwifery" into the Portuguese "*obstetrícia*" [14]. Thus, the word "obstetricians" has been chosen for this article's title because this is what best represents the two kinds of science-educated professionals emerging in Brazil in the early 1800s: the physician and the graduated midwife. Furthermore, since this is not a comparative study, its findings are discussed in the light of the Brazilian historiographical context. Yet, it is expected that an international readership will draw parallels with other parts of the world, especially North America and Europe, given their paradigmatic influence in the studies of gender in general and childbirth assistance in particular. Finally, aware that some arguments may sound unconvincing to readers less familiar with Brazilian specificities, this article ends with a brief discussion of women's role in nineteenth-century Brazil. Hopefully, the Brazilian case—and its unusual respect for women as knowledgeable professionals—may inspire scholars from different countries to explore their own experiences and peculiarities in the history of childbirth.

## 2. Childbirth and Healthcare in Brazil: Now and Then

In the twentieth-first century, despite the expansion of the public health system, traditional midwifery is still part of the childbirth reality in some distant rural communities in Brazil. Women who learn their practice by observing older midwives are in charge of

the care during pregnancy, delivery, and puerperium. In many situations, it is their job to decide whether and when to refer pregnant women to hospitals in the nearest town. Especially in rural Amazonian communities, "they continue to be part of the network of caregivers, where more recently there has been interaction between traditional practices and scientific medical knowledge" [15]. It is impractical to accomplish the total medicalization of childbirth in such a large territory as Brazil, where there are communities that live in remote rural, sometimes forest, areas. Nevertheless, the urban middle-classes—who have universal access to the Unified System of Health (SUS)—tend to be more concerned about the excesses of the medicalized approach, especially the cesarean section, whose rates in Brazil are among the highest in the world. The origins of the urban perspective against the medicalization of childbirth coincide with the 1970's feminist movement, which saw it as a male appropriation of women's bodies and knowledge. In that scenario, reducing medical intervention in childbirth meant to fight against "the oppressive attitudes of paternalistic and misogynistic doctors" [16]. Social science literature still refers to this process as a conflict between "institutionalized male knowledge and female tacit wisdom," represented by the physician and the midwife [17]. Thus, gender is a widespread presence in the scholarship on the process of medicalization of childbirth.

Along with gender, another aspect of the conversation on medicalization in Brazil is the co-existence of traditional practices and scientific knowledge. Throughout the colonial period before the arrival of the Portuguese court in 1808, childbirth assistance as much as healing, in general, were traditional practices. Midwives' and healers' millenary knowledge was hardly informed by the few doctors who came back from medical schools in Europe. Del Priore cites a late eighteenth-century bishop who recognized that it was better to be treated by an indigenous from the backlands who better used his instincts than by a physician from Lisbon. "Health regulations prevented lay people from practicing Medicine, but in the case of colonial Brazil, they were inoperative. Faced with concrete situations, the authorities [fisico-mor] could do nothing against healers' practice [curandeirismo]" [18] (p. 88). When there was no medical training in the country, and few were the individuals educated in European medical schools, the healers were respected by the people and tacitly accepted by the authorities.

For most of Brazil's history, only licensed empirical physicians and barber-surgeons could practice Medicine aside from trained physicians. Still, "they were forbidden to administer internal remedies, which was the privilege of physicians trained in Coimbra." However, the vast and sparsely populated territory discouraged physicians trained in Europe to practice in the country. Thus, given the shortage of trained physicians, the population had to rely on empirical physicians and barber-surgeons who received their licenses after a brief examination by a deputy of the chief physician [fisico-mor] of the Kingdom. Alongside them, there were bleeders, healers, midwives, and apothecaries, who resorted to local medical knowledge of indigenous and African origin. Thus, throughout the colonial period, healing practices resulted from the syncretic combination of European, indigenous, and African expertise.

Before the medicalization process started in the nineteenth century, trained physicians played a secondary role in an environment in which healthcare institutions hardly depended on medical knowledge. The hospitalized assistance was usually carried out in the dependencies of the holy houses of mercy, administered by religious orders. Only at the beginning of the nineteenth century did the first nursing homes [casas de saúde] begin to appear in Rio de Janeiro. They were private establishments owned mainly by physicians trained in Europe and later in Brazil. However, the hospitalization process was not complete. As late as the twentieth century, it was still possible to find cases of itinerant trained physicians practicing in the country's interior. Penna cites the case of the Italian doctor Giovanni Palombini, who between, 1901 and 1914, toured the interior of the state of Rio Grande do Sul practicing Medicine [19].

By the mid-nineteenth century, the scenario of Brazilian Medicine was in an accelerated trajectory of change. Founded in 1835, the Imperial Academy of Medicine would henceforth

defend its project of a medicalized society. It was a time when epidemics and political revolts threatened the court and the provinces, motivating physicians moved by scientific ideology to control public hygiene. At the same time, the medical profession was increasing its direct presence in politics. Physicians equated Medicine with patriotism, trying to show the importance of the nation to have healthy citizens. Then, the axis of influence turned from Portugal, where it had been during the colonial period, toward France. Since it was usual for students to master the French language, they could follow the scientific production in that country.

Nevertheless, the medical project of social intervention only consolidated with the improvement of teaching conditions and the increase in medical intellectual production. From the 1870s onwards, the creation of new publications and the reform of higher education—which, among other things, allowed the entry of women into Medicine—Brazilian Medicine advanced to an international level of reputation. It is worth mentioning that the modernization of Brazilian Medicine, which incorporated the valorization of the scientific method, did not ignore traditional local knowledge. Dr. Nicolau Joaquim Moreira, a member of the National Museum, was an example of this combination of science and popular wisdom. He published numerous articles on the uses of Brazilian plants, including a dictionary of medicinal plants in Brazil published in 1862 [20].

The historiography makes apparent the link between medicalization and the major social changes brought about by urbanization in the nineteenth century. If in other countries the growth of urban settlements followed a sort of gradual process, in Brazil, it was more like a rupture, especially in the case of Rio de Janeiro after the arrival of the Portuguese Court. All of a sudden, in 1808, the colonial town became the capital of an empire. Likewise, the foundation by decree of the two first schools of Medicine marked a rupture with the traditional approach to health care in general and childbirth in particular. However, the rupture did not erase the past, as traditional healthcare approaches persisted throughout the nineteenth century. Arno Wehling notes in Brazilian legal system the same combination of traditional and modern elements as I note in the health system [21]. Writing in 1867, Dr. Moreira de Azevedo still expressed his concerns with the empirical character of Medicine and surgery in the recent past. He doubted that the knowledge of "surgeons and physicians who did not attend classes and only practiced in hospitals" could be beneficial. "Not having the science to properly appreciate the facts and observe the phenomena, symptoms and morbid changes," they were probably no more than "simple nurses and empirical surgeons" [22] (p. 400).

Studying childbirth assistance in nineteenth-century Bahia, Barreto suggests that in Salvador—the second large city of Brazil and home of the first school of Medicine—the technical obstetric culture of the physicians coexisted with the empirical knowledge of traditional midwives. Moreover, she argues that "the scientific culture of birth would only make sense a little later, in the first decades of the twentieth century, based on the alliance between doctors and women" [23]. New cultural and scientific institutions—such as the press, with the first periodicals, the National Library, and higher education institutes like the medical schools—brought new expectations and different values. Despite the resilience of traditional midwifery and healing in everyday life experiences, scientific Medicine became the ideal form of knowledge and practice in the late nineteenth century. It is in this context that we should understand a process of medicalization that has never become total, and a health system that came to accommodate tradition and modernity. Hence, after 1879 when women were granted the right to join the School of Medicine to become doctors, Brazilian health system welcomed women in both extremes —as traditional midwives and as modern physicians. Brazilian in 1879.

## 3. Between "Comadres" and Physicians: The Intermediate Role of the Graduated Midwives

Dr. Fernando Magalhães, "the official historian of Brazilian obstetrics" [24] (p. 174), did not consider traditional midwives (comadres) and obstetricians (physicians and graduate midwives) as part of the same history. This explains why medical discourse stigmatized

most traditional midwives in the late nineteenth century as representatives of ignorance and backwardness. Nevertheless, physicians "were not opposed to the profession of midwives; on the contrary, following a European orientation, they defended the need for midwives who were well informed in obstetric science to attend to natural births and to know how to recognize problems as long as they did not act on their own and call the doctor if necessary" [24] (p. 173). In other words, nineteenth-century physicians were, somehow, asking for graduated midwives to appear. It was as if the physicians needed someone who could deal with women in labor, like a traditional midwife, and have the scientific knowledge about female anatomy and physiology that a physician had.

Two cases reported by Dr. Saboia in 1863 illustrate the partnership between physicians and traditional or graduated midwives. In the first case, a traditional midwife ["comadre"] calls him for help because, after the water bag broke without the birth taking place naturally, a discharge of putrid matter made her believe that the fetus was dead. After saving the baby using forceps, the doctor handed the baby over to the midwife and left. However, hours later, the midwife called him again because the woman was bleeding. Fortunately, the doctor was able to control the bleeding that he said was caused by ignorance of the "comadre" [25] (pp. 141–142). In the second case, the birth assistant is a graduated midwife. The water bag broke, but the midwife noticed that the fetus was not in the normal position. As the contractions had subsided, the midwife would not be able to rotate the fetus. To stimulate contractions, she administered spiked rye. But things did not progress as expected, and the mother was very nervous. Then, fearing that the delivery would take too long, the graduated midwife called the physician. He knew it was necessary to use the forceps. Yet, he only did this after proposing the operation to the parturient and the people present in the room. The baby was born healthy. While the midwife took care of the newborn, the physician worked to stop the mother's bleeding that followed the delivery. After that, the physician remained at her bedside for one hour, waiting for the placenta to be expelled. He then left, recommending that a friend of the mother watched her [26] (pp. 162–163).

These medical cases reveal significant similarities and distinctions between traditional and graduated midwives that suggest why physicians might have been pleased by sharing their practices with women who understood the science of childbirth. Firstly, both the midwives only called the physician when the labor became too complicated for them to handleby themselves. This was to the advantage of the general practitioner, as, while the midwives assumed the long-hour job, they freed up the physician's time to attend to more lucrative types of medical calls (In the nineteenth century few were the physicians that specialized in only one field). Secondly, no matter how trained the midwives were, they were the ones in charge of taking care of the newborn. This is another way in which their work liberates the physician so that he could take higher-skilled jobs.

Apart from these similarities, the distinctions between the two kinds of midwives make the existence of the graduated one even more beneficial for physicians. The main difference between them appeared when labor did not progress naturally. The traditional midwife did not have any resources but call for help. On the other hand, the graduated midwife knew how to rotate a fetus and which medication to use to stimulate contractions. She only called for help because of the emotional state of the parturient. Contrarily to the traditional midwife in the first case, who was blamed for the complications and was called ignorant, the graduated one showed confidence and was praised by the physician. The more skilled a midwife was, the more she contributed to the delivery's success. Therefore, skillful midwives prevented physicians from being called just to extract a dead fetus. Dr. Saboia reports also reveal interesting aspects of the physician's behavior. Although he knew his presence was requested because of his techniques and instruments, he refrained from using the forceps without the consent of the parturient. Besides, he remained at the parturient's bedside for one hour after the delivery, just like a midwife would do. Thus, analyzing these cases, where feminist scholars may see power struggle in hierarchical relations, I see pragmatism in a dynamic society.

The promulgation of the law of 3 October 1832 would respond to the implicit demand for skillful and science-trained midwives. It made the Medical Schools of Bahia and Rio de Janeiro responsible for granting the certificates required for the exercise of Medicine, pharmacy, and midwifery, as well as for verifying the certificates issued by foreign institutions [5]. Mme. Durocher, whose work is studied in this essay, completed the two-year course on midwifery at the Medical School of Rio de Janeiro in 1834. She was born in France but moved to Brazil with her family when she was a child. In the same year, Durocher registered her naturalization request and her certificate of midwifery, becoming the first Brazilian graduated midwife [10]. "The idea of midwives with a certificate granted by doctors began to spread. Graduated midwives became the most legitimate and requested by the most powerful and civilized families. It was at that point that French midwives trained in their country also arrived in Rio de Janeiro, bringing new techniques and prescriptions" [6] (p. 73). This does not mean, however, that French midwives would be safe from criticism. Mme. Durocher warned not only about the ignorant "comadres" but also the "charlatans" disguised as graduated midwives [27] (p. 293). Decided to show the defects of the obstetrics training provided by the medical school, Mme. Durocher asserted that "most foreigners easily obtained the dispensation of preparatory tests." Consequently, women "who did not know anything" were enrolled in medical school. "Without any basis of instruction, our school accepts women whose purpose is not to conscientiously learn a science [...], but rather to reach a document called a diploma that enables them to earn money by receiving children and often taking them to the foundling wheel" [27] (pp. 290–291).

As the term "midwife" [parteira] can be used to refer to both traditional [comadre] and graduated women who assist childbirth, the two types may be confounded when it comes to criticism and praise from physicians. For example, in 1863, a physician suggested that the midwife was guilty of a case of rectovaginal fistula. But how to know if she was a traditional or graduated one? The report informs that the parturient lived outside the city ["moradora fora da corte"] [28] (p. 97). Therefore, we can assume that it was a traditional midwife since the few graduate midwives worked in the urban centers. Another issue of the *Annaes* reported a case where the parturient "called, to assist her, a very intelligent Brazilian midwife, Ms. Tygna" [26] (p. 162). Here, the fact that the physician called the midwife by her name made clear that she was a graduated one. It would be a valid hint even if the content of the report were not complimentary.

Some authors have implied that competition motivated the physicians' criticism toward midwives. Based on advertisements in the *Jornal do Commercio* newspaper between 1831 and 1900, Medeiros et al. have claimed that physicians and graduated midwives competed against each other in the childbirth market [29]. I agree that, especially when the scientific appeal of Medicine became more apparent in the 1870s, the two professions came to occupy the same urban upper-middle-class niche. However, I do not see this as evidence of competition. In fact, a close reading of the *Annaes Brasiliensis de Medicina of the Imperial Academy of Medicine* leads me to believe that the relationship between physicians and graduated midwives was more characterized by partnership than competition. As the two cases reported by Dr. Saboia demonstrate, and Mme. Durocher's words suggest, only the traditional midwives were criticized for their ignorance. The graduated midwives, on the contrary, deserved praise. It is necessary, therefore, to distinguish the kind of midwife at whom the criticism aims.

Other medical cases presented to the Academy by Dr. Saboia confirm the respect and admiration toward graduated midwives and indicate how they played complementary roles in the delivery scene. His communication of 19 October 1863 is particularly revealing, as he politely refers to three midwives by name in the same text. He starts by telling the case of a patient of the "intelligent" Ms. Tygna [Thereza Jesuina Tygna] and the progression of complicated labor that was initially perceived as normal [30]. Tygna medicated the parturient with spiked rye to accelerate the contractions, but she was afraid that it could take too long, putting the fetus at risk. The midwife called Dr. Saboia to come to assist. As

the uterus remained inert, he decided to use the forceps and successfully delivered a healthy baby boy, of whom Ms. Tygna took care [26] (p. 162). Dr. Saboia chose to present that case to reflect on the combined use of forceps and spiked rye—a medication to stimulate uterine contractions. In this regard, he referred to a previous situation where the midwife in charge was Mme. Durocher. As she had argued that it would be good to wait for the medication to act, he used the forceps too late, and the child was born in asphyxiation (fortunately, the child recovered). Mentioning that he had experienced a similar situation with Ms. Gaullier as the midwife in charge, he claimed that the forceps should be used soon if the waters had broken for six hours and the uterus remained inert [26] (p. 163). This passage shows that the physician, Dr. Saboia, who was already a member of the Imperial Academy of Medicine, did not feel entitled to question the midwives' decision until he had enough experience with the medication they used. He only made prompt use of the forceps with Ms. Tygna's patient. In the other two previous cases, he did what the midwives suggested. In my view, such behavior implies partnership as much as a negotiated authority, which seemed very much based on knowledge and experience—not gender.

The relative symmetry between graduated midwives and physicians in the late nineteenth century is also expressed in the fact that both were supposed to medicate patients. In criticizing the medical schools of Brazil, Dr. Gama Lobo, editor of the *Annaes*, defended the necessity of the midwives to learn therapeutics and anatomy so that they could medicate and use the forceps properly. He did not assume those skills should be a monopoly of male obstetricians. In the following month, Gama Lobo recognized that Mme Durocher was a midwife who had "knowledge of anatomy, pathology, and therapeutic physiology," praising her as "the first midwife of Brazil." This happened years before she was admitted to the Imperial Academy of Medicine [31]. Note that the *Annaes Brasiliensis de Medicina* published contributions of midwives even if they were not members of the Academy. Note that Mme. Durocher frequently published before being admitted and that Ms. Tygna, in 1865, discussed a case of "Congestion of the uterus causing abortion" [32].

## 4. Durocher and Tygna: Two Graduated Midwives in 1860's Rio de Janeiro

Mme. Durocher and Ms. Tygna were two respected midwives who happened to publish their works in the *Annaes Brasiliensis de Medicina*. Thus, their writings, which are available to digital access at the Brazilian National Library [Biblioteca Nacional], are important primary sources for the study of midwifery practices in the nineteenth century. Through their perspectives, one can have a glimpse of contemporary debates and the social relations inside the delivery scene, where graduated midwives and physicians were frequent presences alongside the mother and the newborn child. Thus, I examine Ms. Tygna's report on uterus congestions provoking abortion, presented in 1864 [32], and Mme. Durocher's considerations on the use of spiked rye, presented in 1866 [33]. In both texts, it is clear that rye was a popular medication among midwives and physicians. Its activity in promoting uterine contractions raised concerns about the use of rye as a cause of miscarriage/abortion (Note that, in Portuguese, both terms "miscarriage" and "abortion" are translated as "aborto." Although "miscarriage" implies spontaneity, the texts examined in this research do not use any adjective to qualify the word "aborto." Thus, here, although I try to use the English words as they best suit the context, sometimes the meaning is ambiguous). As Mme. Durocher explained, "when the rye acts upon the circulatory system, it is a powerful hemostatic and hyposthenic agent." In that case, it slows the pulse. However, "if the rye acts upon the nervous system, it can excite uterine contractions." Since rye's effect depends on many factors, especially concentration, Mme. Durocher warned that it is an "infidel" medication, "on which one must not or cannot count," and its use can be deadly both to mother and child [33] (p. 46).

Ms. Tygna questioned if rye could cause contractions when the uterus is not prepared for it. To support her reflections to the Academy, she presented a miscarriage case of a parturient in the third month of pregnancy. After examining the patient and judging the miscarriage as "inevitable," the midwife prescribed "potion of Velpeau" [rye preparation].

As a result, the bleeding stopped, and the patient was feeling well. However, a few days later, the bleeding restarted. Ms. Tygna reported all the previous procedures to the family's physician, who "agreed with everything that I [she] have done." As she believed that "a uterus congestion had been the cause of the hemorrhage," she suggested the "application of leeches, which he opposed." Instead, he advised her to repeat the potion of Velpeau, and she "gave in." The pregnancy went on, and the patient gave birth without any accident. "What could I conclude from this observation?", asked Ms. Tygna [32] (pp. 66–68). Ms. Tygna hypothesized that the rye potion had worked solely as a hemostatic agent, controlling the bleeding and, consequently, it interrupted—it did not provoke—the miscarriage. Two years later, Mme. Durocher would comment on her colleague's report: "Many times I used Velpeau's obstetric potion to complete the expulsion of the egg and, instead of that result, I saw the hemorrhage stop and the pregnancy continue its course. Midwife Tygna had presented in this journal an identical observation." Mme. Durocher then explained this phenomenon as due to the immaturity of the uterine muscles in the early stages of pregnancy [33] (p. 46).

Thus, both women seem to agree on the protective action of rye against miscarriage when used to control bleeding in the early stages of pregnancy. Yet, Mme. Durocher stressed the dangers of rye by midwives "for whom science is nothing and gain is everything" [33] (p. 47). This quote expresses Mme. Durocher's criticism against the midwives who obtained certification in questionable ways and who were willing to offer abortive solutions in return for money. In this context, she mentioned the "miserable mothers who confessed to having taken rye to no avail at different times in order to provoke abortion and hide their dishonor" [33] (p. 48). Considering that Brazil is a Catholic country, it is interesting to notice how little embarrassment the discussion of abortion and women's sexuality caused, at least in the medical literature of the late nineteenth century. An illustrative example is Dr. Saboia's observation of a case of miscarriage, published in the *Annaes* in December 1863. It happened to a married couple who, having already three children, made use of an "innocent" sexual maneuver "that prevents the sperm from entering the vagina" as a contraceptive method. As this strategy is not safe, the wife became pregnant, and after three months, Dr. Saboia was called for an eventual miscarriage [34] (pp. 165–166).

What makes Dr. Saboia's report relevant to this research is the kind of rhetoric he used and the explanation he gave for the cause of miscarriage, which "had occurred before that lady could take the medication I [he] prescribed." The physician affirmed that "there was no accident or moral distress" to justify the event. More importantly, he emphasized that the woman was "adored by her husband and she enjoys [ed] all the imaginable fortune in the world, and she deserves [d] it" [34] (p. 165). Thus, after protecting the woman's reputation, Dr. Saboia presented his hypothesis for the cause of the miscarriage: frustrated orgasm. He built his argument around the sacred function of reproduction that starts with copulation – "this act surrounded by ineffable and sublime pleasure for both agents." He argues that "if the sensual pleasure does not declare itself to the woman when she is prepared for it, and the penis is removed, the uterine orgasm continues and the uterus becomes sensitive and predisposed to congestion" [34] (p. 166). By this logic, Dr. Saboia arrived to a possible mechanism to explain the phenomenon that Ms. Tygna observed: "congestion of the uterus causing miscarriage" [32] (pp. 66–68).

Thereza Jesuina Tygna and Marie Josefina Mathilde Durocher were among the most reputable midwives in Rio de Janeiro, whose relationship was marked by friendship and professional admiration. In 1870, the newspaper *Jornal do Commercio* informed readers that Ms. Tygna moved to the house where Mme. Durocher had lived [29]. In February 1871, just before becoming a member of the Academy, Mme. Durocher praised Ms. Tygna in her article titled "Deve ou não haver parteiras?" [Should there be midwives or not?]. She pointed out that Ms. Tygna had "a doctor library" [uma biblioteca de medico]. With that phrase, Mme. Durocher synthesized her vision of the ideal midwife—someone who is entitled to assume the responsibility for the lives of the woman and the child due to her

scientific knowledge [27] (p. 292). In this article, Durocher prophesizes the transformation of the graduated midwife into the female physician.

## 5. Medicalization of Childbirth: Misogyny or Scientificism?

*Deve ou não haver parteiras?* [*Should there be midwives or not?*] is a fundamental source to understand the role that a graduated midwife was expected to play in late nineteenth-century Brazil. The long article was written by Mme. Durocher, in 1871, six years after Dr. Gama Lobo argued that midwives needed to learn therapeutics and anatomy. Thus, Mme. Durocher's commentary is part of a conversation that would lead to the 1879 reform of higher education. The new law that opened medical schools to women also justified the reforms conducted by Dr. Vicente Saboia. He was at that time Director of the Medical School of Rio de Janeiro. Among the reforms, there were curricular changes and also more prosaic adaptations such as making restrooms available for women. In the context of the projects to reform the medical school that would lead to "the golden period of national medicine" [6] (p. 71), Durocher advocated for a three or four-year course of midwifery. She lamented that the midwifery students had only two years to learn content taught to medical students during four or five years. She recognized that such a change would reduce the number of midwives available. However, she believed that the higher qualification of the few who graduated would justify the decision. In addition to the good it would mean to society, they could help "destroy the prejudice of doctors, which is supported by these ignorant "comadres" and charlatan-midwives [graduated]" [27] (p. 293).

In the article, Durocher appeals to the trope of the nineteenth century as "a century of progress" to advocate for a longer, scientific-based course of midwifery [27] (p. 294). In so doing, she distanced the figure of the graduated midwife from the traditional women who had assisted childbirth for immemorial times. Her attitude resembles that of Dr. Moreira de Azevedo mentioned above, who distanced modern physicians from the ancient healers and empirical surgeons. As the end of the century approached, it was even harder to deny the influence of scientificism on health practices. By phrasing its title as a question, she expressed a dilemma that had grown stronger as the role of a midwife resembled that of the physician: Should there be midwives or not?

It is important to consider that, by the end of the nineteenth century, there was a gigantic difference between graduated and traditional midwives regarding their knowledge and social role. The close reading of the *Annaes Brasilienses de Medicina* shows that the place that midwives such as Mme. Durocher and Ms. Tygna occupied in the family dynamics of childbirth had little resemblance to the place of the traditional midwife or "comadre." This word, borrowed from kinship terminology, suggests that the traditional midwife is part of a network of female solidarity involving "gift, kinship, affection" [15]. Since the moment of her graduation in 1834, the first midwife of Brazil had worn a gendered outfit to express an image of professionalism and differentiate herself from the domesticity of the "comadres." The descriptions and portraits of Mme. Durocher draw much attention to her appearance, which a contemporary described as "an ill-defined mix of man and woman" [6] (p. 95). In an autobiographical account at the final part of her article, Mme. Durocher explained her decision to dress like a man: "I thought that this exterior of mine should act a lot on women's morale, inspiring more confidence and distinguishing midwives from ordinary women" [27] (p. 299). Thus, in one stroke, she distanced herself from the traditional midwives and the female ethos.

Maria Lucia Mott noted that "although wishing to serve as a model for the graduated midwives, it seems that the masculine costume adopted by Mme. Durocher was never copied; on the contrary, she was even ridiculed" [10] (p. 116). Nevertheless, the intellectual demand of the midwifery course offered in the Medical School of Rio de Janeiro brought women and men physically close together. Even though the female students were not required to dress like their male classmates, earnest midwifery students like Ms. Tygna mingled with medical students to learn the scientific foundations of their future profession. Mme. Durocher commented on the willingness of the medical students [all male by then] to

teach lessons of anatomy to their female classmates and "to give them all the explanations they needed about physiology, therapeutics, and hygiene" [27] (p. 292). Hence, I argue that the graduated midwife played the role of a historical intermediate in the process of medicalization of childbirth that occurred during the modernizing nineteenth century. Originally emerging from amongst women who assisted childbirth in the communities, the midwives progressively differentiated themselves from their traditional counterparts. Such differentiation resulted from an increasing effort to institutionalize and control the millenary craft of midwifery—as it has happened with many other crafts in late modernity. However, since this institutionalization was guided by scientific education, the graduated midwives tended to become more like physicians. Thus, once the transition had completed, what role would rest for graduated midwives? For some few decades, one difference still held. Midwives were female, and physicians were male. But once women become doctors, what difference would gender make?

The first Brazilian female physician graduated in 1887 at the Medical School of Bahia. Soon other women would receive their degrees in Medicine. The historiography has registered that Rita Lobato Lopes was warmly welcome by professors and classmates, who "invited her to parties, balls, and walks" [24] (p. 243). Her social life, however, did not prevent her from studying. Dr. Rita Lobato completed her course and received her doctoral degree with the thesis "Parallels between the Methods Recommended in Cesarian Operation" [Paralelos entre os Métodos Preconizados na Operação Cesariana] [35]. Remarkably, the first female doctor chose to study, in 1887, a topic at the center of the debate on the medicalization of childbirth nowadays: C-section. Although the high rates of C-section in the present have been very controversial, it is important to bear in mind that, now and then, this is a procedure meant to save lives, mother's and children's. That was certainly the motivation for Dr. Rita Lobato to have chosen it as a topic for her thesis in 1887. I suggest that the high rates of C-section in Brazil result, at least in part, from the belief that the physician, male or female, advocates for the woman's interests. As the Brazilian medical class refrained from expressing misogynic opinions, physicians are regarded as women's protectors, not oppressors. Hence the process of medicalization of childbirth went on and became even stronger as more women became physicians themselves.

## 6. Brazil: An Unusual Case?

The history of childbirth in nineteenth-century Brazil has shown the emergence of new characters on a stage previously populated only by women. Across the world, they had been guided by traditional knowledge, orally transmitted from mother to daughter, aunt to niece, neighbor to neighbor across generations. Men would have to wait until the scientific revolution began to transform social values to be allowed in the delivery room. It happened in different rhythms, depending on the place. In England, for example, it was over the course of the eighteenth century that the presence of male medical practitioners became more common. But even then, "midwives continued to oversee births, with surgeons being called only where a surgical intervention was required. Even if inside the hospitals, practice did not immediately change radically" [36] (p. 326). In Brazil, the medicalization of birth began in the early nineteenth century. But unlike in England, where male doctors were responsible for introducing scientific knowledge to help labor, the Brazilian system of higher education created two kinds of obstetricians. One was the male physician, and the other was the female graduated midwife. Both were scientifically educated and regarded the traditional midwives with distrust and contempt.

This tripartite model is crucial to understand how it was possible that graduated midwives were well respected by male medical practitioners, as the sources have shown. Possibly under the influence of North American and European works, it is not rare that the evident contempt that physicians had against traditional midwives in Brazil is interpreted as gender prejudice. Indeed, where all the midwives were unskilled women and all the physicians were skilled men, it is easy to take a simple correlation between gender and status as if it was actual causation. However, it was not the case in Brazil, where a third part–

the graduated midwife–presented the dual characteristic of being woman and educated. More significantly, as early as the 1830s, this kind of female professional received the same training as male physicians interested in obstetrics. They sat side by side in the same classrooms in the Medical Schools of Rio and Bahia, where the country's best professors taught both groups, tuition-free. Then, when, in 1879, women could enter the medical school for the five-year program of Doctor in Medicine, they already had a long history of coexistence and friendship with male colleagues. Thus, the study of the dynamics between traditional midwives, graduated midwives, and male (and then also female) physicians confirms that the devaluation of the first was not a matter of gender. As it was noted for the English context, "this was part of a broader trend that valued scientific knowledge and clinical observation" [36] (p. 334).

Still, significant differences appear when comparing obstetricians' training in Brazil with England and the United States in terms of gender. In late nineteenth-century England, midwives' practical training happened "in lying-in hospitals . . . . for a fee" [36] (p. 326). Contrarily, in Brazil, they received scientific training at the medical school, alongside physicians and surgeons, free of charge. Thus, although in England "midwifery training was aligned with medical education more generally" [36] (p. 326), it was not as much as in Brazil. Moreover, even after the Act of 1902, British midwives still were "unqualified persons," and because of that, they were "only certified for attendance upon normal labour" [37]. Brazilian graduated midwives had been trained to perform the same procedures as male obstetricians, including prescribing drugs. As a result, instead of evolving to become nurses, Brazilian graduated midwives gave birth to female-physicians.

Looking at the international context, it is a fact that female-physicians appeared first in the United States. In 1849, three decades before Brazilian universities were opened to women, "Elizabeth Blackwell graduated from Geneva (New York) Medical College, the first woman to receive a regular medical degree in the United States" [4] (p. 266). But in contrast with the 5-years, coeducational medical education in Brazil, American women studied for only two years (the same duration of obstetrics courses for graduated midwives in Brazil) and in a female-exclusive environment, 'protected' from male colleagues.

"The inferior education offered to many women graduates of female-only proprietary medical schools" displeased Blackwell and motivated her to establish "a high-standard, rigorous 4-year medical school at the New York Infirmary" [38] (p. 468). But even recognizing "that positive relationships with men would provide support, guidance, and legitimacy to their work," female-only medical schools would be a reality until the turn of the twentieth century in the United States [38] (p. 266).

In addition to these institutional differences, which impacted education and professionalization, Brazil stands out in cultural and societal terms. The Victorian moral standards have never taken roots in Brazilian soil. Unlike North America and Europe in the nineteenth century, the country experienced urbanization without the phenomenon of industrialization. The "cult of domesticity" helped the Victorians to cope with the tension and anxiety caused by the rise of industrial capitalism [39]. Therefore, a rural country where towns and cities had grown due to gold mining or transatlantic trade did not feel similar tension. In fact, throughout the nineteenth century, the so-called "two separate spheres" remained quite entangled in most parts of Brazil. Business and political activities took place inside the homes, where women were expected to contribute to the reputation and success of the family. In the countryside, patriarchal families administered their fortunes from their communal residence in the plantations. In the towns and cities, it was not rare to find women in charge of businesses, especially "where male economic activities required a certain absenteeism" [40] (p. 179). Commercial companies were usually located on the ground floor of the owners' home in Rio de Janeiro. As a historian notes, "The architecture of the townhouses [sobrados] developed making the street 'a servant of the house'" [41] (p. 224). The frontier between street and home was porous, and men were part of the domestic life as much as women had access to the public world. There was no sharp divide between public and private lives comparable to industrialized countries. Only by

the turn of the twentieth-century modernization would transform the relation between home and street. New regulations restricted music, festivities, and social gatherings in the public space, dissolving old forms of solidarity and introducing values of a kind of sociability centered in the nuclear family. Only then "women had to learn to behave in public" [41] (p. 228).

A quick observation of women's education in Brazil helps to illustrate the pragmatism that distanced the South American country from the moralism of industrialized (Protestant) countries in the nineteenth century. In 1827, Senators discussed an amendment in the law regarding primary school. Some of them defended the argument that girls should receive instruction in Arithmetic since "many women married to merchants had lost their businesses after becoming widows, for lack of this instruction" [42] (p. 582). Even those who opposed it—and by the influence of "the sages of Europe" believed that it was enough to teach the girls what was useful for the domestic economy—agreed with the teaching of sciences "to the daughters of rich and noble people" [42] (p. 582). Soon, the curriculum of the new schools for girls' education would not seem that different from boy's schools. The *Collegio Augusto*, founded by the feminist Nisia Floresta, taught History, Geography, and also Arithmetic to girls [43]. In a time when progress was a national goal, an article signed with the pseudonym "Propagator of Industry" defended a "convenient education" for women. He argued that "woman makes the fortune or the impoverishment of her husband and posterity" [44]. Seventy-two years later, the Senate would discuss a project of law guaranteeing the exercise of liberal professions by women. According to a newspaper, the senators unanimously rejected the project. They claimed that "the Constitution does not impose the least restriction to the exercise of any profession by women" [45]. Indeed, in the 1890s, one can find news like the one that appeared in the newspaper *O Paiz* on 31 March 1891: "Antonieta Dias, talented [female] doctor in medicine, ... is establishing her clinic of obstetrics and gynecology" [46].

It is in this context that one should interpret the words of mid-nineteenth-century medical authorities that valued both male and female midwives, provided they were "well instructed." In 1835, Dr. Sigaud predicted that "very soon we shall possess worthy and skilled male midwives [parteiros] and instructed female midwives [parteiras] who will populate the capital and the provinces" [47] (p. 36). Apparently, he saw no impediment for offering each other the kind of training they traditionally lacked—for physicians, the practical knowledge of the female body, and for midwives, scientific knowledge in obstetrics.

## 7. Conclusions

The Museum of the Nacional Academy of Medicine, in Rio de Janeiro, exhibits a sculpture in bronze titled "struggle of physician against death" [A luta do médico contra a morte] [24] (p. 139). As Martins observes, "in Brazil, we have no testimony of a negative reaction to the admission of physicians in the delivery scene" [24] (p. 174). There is no sign of reaction against women in Medicine either. The reason for such a gender-blind environment may lie in the fact that obstetrics, as a medical specialization, evolved from a history where physicians attended only cases of complicated deliveries. As they might have saved more lives than they faced death, their image as protectors of life solidified. At the same time, as women committed themselves to education, side by side with male students in the medical school, they soon proved their capacity to be not only graduated midwives but also physicians.

Considering that the nineteenth-century culture praised science so much, a doctoral degree in Medicine might have seemed as heroic as saving lives. The attentive reading of *Annaes Brasilienses de Medicina* showed me professionals—men and women—who diligently worked to improve the health and wellbeing of women and their children. The ways and strategies they used reflected a combination of healing traditions and modern science in ebullition in the late decades of the Brazilian Empire. So, what difference did gender make? I conclude that, since the graduated midwife was a woman of science, she was able to fill

the gap that isolated the female sphere of care from the male sphere of science. Once the bridge was paved, men and women conquered the right to choose which direction to go.

**Funding:** This research received no external funding, except for the graduate stipend paid by the University of Rochester to the author.

**Conflicts of Interest:** The author declares no conflict of interest.

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
