# Peer review of "What Difference Gender Makes: Obstetricians in Nineteenth-Century Brazil"

_2409-9252, doi:10.3390/histories1040022_

Round 1
Reviewer 1 Report
Comments and Suggestions for Authors
The author is to be congratulated for an excellent, well-argued and carefully researched article, which makes an important contribution to the historiography of childbirth, not only in Brazil, the focus of the case study, but internationally. The author challenges the view of history as gender politics, so prevalent in much of the writing on the history of obstetrics and childbirth in particular. In much of the latter writing, both women and midwives are seen as victims of a power play by male doctors as childbirth was medicalised. By contrast, based on his or her own research, this author argues, ‘where feminist scholars may see power struggle in hierarchical relations, I see pragmatism in a dynamic society.’ The sources are the Annaes Brasiliensis de Medicina – the official publication of the Imperial Academy of Medicine in Brazil, with a particular focus on two midwives, Thereza Jesuina Tygna and Marie Josefina Mathilde Durocher. The author explores the relationship between these ‘graduated’ midwives and male doctors, and argues that there was respect and collaboration, not competition, which eventually eased the way to the admission of women to medical school. The conclusion that the medicalisation of childbirth should be viewed as less about misogyny and more about the embracing of science by both men and women, is well supported in this article.
Author Response
Dear Reviewer
Thank you so much for so positive comments on my manuscript.
Please, find attached the cover letter for the revised version.

Reviewer 2 Report
Comments and Suggestions for Authors
Thank you for giving me the opportunity to review this paper exploring the gendered development of maternity care in 19thc Brazil.
The topic is a really interesting one as it has the potential to add to our understanding of the development of obstetric 'science' in the 19thc across a range of cultures and countries; the majority of work thus far has been in the North American and European context, so this work has the potential to add breath and depth to our understanding of the issues.
The paper is generally well written and makes use of an interesting and relevant primary source in order to ground the arguments. However there are a number of significant weaknesses which impact on the usefulness of the paper in its present state.
- it would be helpful to be much clearer about language. for example, the title talks about 'obstetricians' which in the western context has a very specific meaning as a medically trained individual; and the word was not current until the 20thc. if you are using it specifically in the Brazilian context to cover both drs and graduated midwives then you need to be clear about this; how and when it was used and for whom. otherwise it is confusing for the reader. You need to be clear about the titles you are using throughout your work.
- For an international readership you need to be really clear about the different types of practitioner; who they were, where they worked and what the training looked like. My reading of your evidence is that that the role of graduate midwives is analagous to trained midwives in eg England after 1902 but more context would make this clear. what type of women became graduated midwives, how did they fund their training etc.
- You seem to make the point that graduated midwives were well respected by medical practitioners etc. this is unusual in the international so it would be useful to explore critically how and why you think this was - was it the general circumstances of the culture and role of women in Brazil?
- Your work needs a greater historiographical context; you mention Leavitt's work but that is all. Wider reading would allow you to be clear about continuities and disconnects in the Brazilian situation.
- You mention Scot's work on gender as your organising feature, but gender is only explored in a superficial way and i wonder if there are other factors that you miss; particularly social class as often the battle seems to be between drs and graduated midwives on one hand and traditional attendants on the other. it would be good to explore this aspect in more detail. is it about gender, or about class and education regardless of gender?
- section 2 is interesting but it is not clear how it links to your argument. think really carefully about your main point and make sure this is threaded through your discussion. for example in this section you discuss urbanisation, which may well be another important factor, but then this is not picked up again through the paper.
- the story in lines 251-268 is interesting but you have not really critiqued it; This doesn't seem to be about gender but about culture and education - and possibly again about class. it would be helpful to explore the nuances of this in greater detail
Overall this paper needs work; grater attention to the literature and a more nuanced understanding of the material you are presenting and the story it tells
Author Response
Dear Reviewer
Thank you for your detailed comments and for the opportunity to revise the manuscript.
Please, find attached the cover letter of its revised version.
Sincerely,

Reviewer 3 Report
Comments and Suggestions for Authors
The article is highly interesting and addresses a much-debated issue, but I am not fully convinced that the medicalisation of midwifery was as smooth as the source materials of the text indicate and that trained midwives were treated as fully equals by medical science. As far as I can see, a deeper source critical discussion would have been helpful since the Annales are a particular kind of genre that follows particular writing conventions that likely conceal the dimension of power. The analysis on lines 341-364 reveals that midwife's work is judged from a doctor's perspective and may be biased.
It is important to critically scrutinize previous research, even gender and feminist scholarship, but it is equally important to make sure that the criticism is well-grounded. At several occasions the author fails to convince me as a reader since they do not discuss the matter at hand in depth (e.g. lines 293-4; 328-334 are a good example of not deepening the analysis).
Author Response
Dear Reviewer
Thank you so much for your comments and for the opportunity to revise the manuscript.
Please, find attached the cover letter of its revised version.
Sincerely,

Round 2
Reviewer 2 Report
Comments and Suggestions for Authors
Thank you for allowing me to review the revised version I feel that this is much stronger because it places your topic in much broader context which in turn strengthens your arguments